# Lightweight Machine Learning Method for Real-Time Espresso Analysis

Jintak Choi [1], Seungeun Lee [1], Kyungtae Kang [2] and Hyojoong Suh [3,*]

1  Department of Applied and Computer Science and Engineering Major in Bio Artificial Intelligence, Hanyang University, Ansan 15588, Republic of Korea; jtchoi@hanyang.ac.kr (J.C.); seungeunlee@hanyang.ac.kr (S.L.)
2  Department of Artificial Intelligence, Hanyang University, Ansan 15588, Republic of Korea; ktkang@hanyang.ac.kr
3  School of Computer Science and Information Engineering, The Catholic University of Korea, Bucheon 14462, Republic of Korea
*  Correspondence: hjsuh@catholic.ac.kr

**Abstract:** Coffee crema plays a crucial role in assessing the quality of espresso. In recent years, in response to the rising labor costs, aging population, remote security/authentication needs, civic awareness, and the growing preference for non-face-to-face interactions, robot cafes have emerged. While some people seek sentiment and premium coffee, there are also many who desire quick and affordable options. To align with the trends of this era, there is a need for lightweight artificial intelligence algorithms for easy and quick decision making, as well as monitoring the extraction process in these automated cafes. However, the application of these technologies to actual coffee machines has been limited. In this study, we propose an innovative real-time coffee crema control system that integrates lightweight machine learning algorithms. We employ the GrabCut algorithm to segment the crema region from the rest of the image and use a clustering algorithm to determine the optimal brewing conditions for each cup of espresso based on the characteristics of the crema extracted. Our results demonstrate that our approach can accurately analyze coffee crema in real time. This research proposes a promising direction by leveraging computer vision and machine learning technologies to enhance the efficiency and consistency of coffee brewing. Such an approach enables the prediction of component replacement timing in coffee machines, such as the replacement of water filters, and provides administrators with Before Service. This could lead to the development of fully automated artificial intelligence coffee making systems in the future.

**Keywords:** machine learning; color clustering; GrabCut; coffee crema; espresso





## 1. Introduction

Coffee is a globally beloved beverage, and achieving proper espresso extraction is essential to delivering the desired taste and quality in a cup of espresso. One crucial element in evaluating espresso is the coffee crema, the golden brown foam that forms on top of the espresso shot. The crema serves as a visual and sensory indicator, contributing to the overall espresso experience. To unlock the full flavors and characteristics of the coffee beans, achieving an optimal extraction becomes imperative. While there are existing standards such as the Specialty Coffee Association of America (SCAA) [1–3] and Agtron M-basic machine [4,5] for evaluating espresso extraction, the assessment primarily relies on human visual inspection. However, this reliance on subjective human judgment presents inherent challenges. Inconsistencies and variations in the evaluation process make it difficult to obtain reliable and standardized results. Therefore, there is a pressing need for objective and reliable methods to evaluate coffee crema in coffee experiments, ensuring more accurate and consistent evaluations of crema quality.

In recent times, the integration of machine learning (ML) techniques and computer vision has garnered significant interest as a promising solution across various applications [6,7]. However, despite the notable advancements in ML and computer vision, there remains a lack of established standards and limited efforts to connect these technologies with the coffee industry. Although ML techniques demonstrate immense potential, their widespread adoption in the coffee industry is hindered by the lack of standardized methodologies and seamless integration with coffee machine systems. Furthermore, while ML shows promise, its practical application on real devices has encountered certain limitations. This can be attributed to factors such as computational constraints, memory requirements, and energy efficiency concerns. With the increasing demand for immediate results and on-the-spot decision making, there is a pressing need for real-time crema analysis in the coffee industry.

The emergence of AI-enabled Jetson Nano computers allows the integration of machine learning into resource-constrained devices and Jetson Nano systems, enabling the implementation of machine learning models on actual devices. Specifically designed lightweight [8], efficient algorithms have been developed to accommodate the limitations of edge devices. These remarkable advancements in edge computing, coupled with the integration of machine learning, have paved the way for the practical application of machine learning on real devices.

In response to these challenges, we present a lightweight ML algorithm for extraction judgment that leverages crema images extracted from coffee machines. Unlike existing methods, which lack Agtron-based analysis for actual extracted results, we propose a novel analytical approach using unsupervised learning through clustering algorithms. Our proposed algorithm incorporates a two-step ML approach by combining the Grabcut and Clustering algorithms. To ensure efficient and timely judgment, we seamlessly integrate this algorithm into the hardware of the coffee machine, enabling real-time assessment during the brewing process. Our approach is based on the premise that the optimal extraction model can be determined by analyzing the espresso crema image captured from a commercial coffee machine. Additionally, the intelligent system continuously monitors the condition of the beans and adjusts the motor control in the coffee grinder to provide precise and consistent grinding results. Real-time data analysis enables dynamic adjustments to achieve the desired extraction.

Our contributions are as follows:

- Proposingl a lightweight algorithm for real-time analysis of coffee crema extraction.
- Creating an intelligent automatic crema extraction system in the coffee machine through a Jetson Nano single board.
- Providing users with ML-based coffee-machine-monitoring control service.

This study discusses approaches to ensuring the quality of services, including service-level agreements in coffee service, monitoring, and adaptation. These mechanisms enhance the reliability, performance, and availability of services.

When discussing the delightful essence of coffee, there are various crucial elements with which we may not be familiar. Temperature, pressure, extraction time, grind size, extraction quantity, bean aging, and more collectively contribute to the overall result known as espresso crema. When these requirements are met, as shown in Figure 1, espresso crema can be broadly categorized into three distinct forms [9]. To obtain the ideal image (color) representation of espresso crema, Figure 2 describes the procedure for providing users with visualized data using machine learning from artificial intelligence. The data is preprocessed with Grabcut, and Color Clustering algorithms are used for analysis, offering users a visual representation of the espresso extraction method.

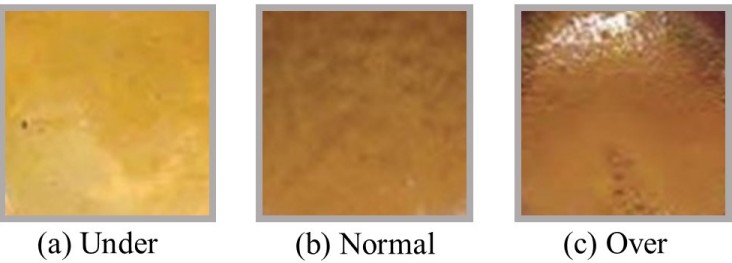

(a) Under      (b) Normal      (c) Over

**Figure 1.** Espresso crema in 3 forms.

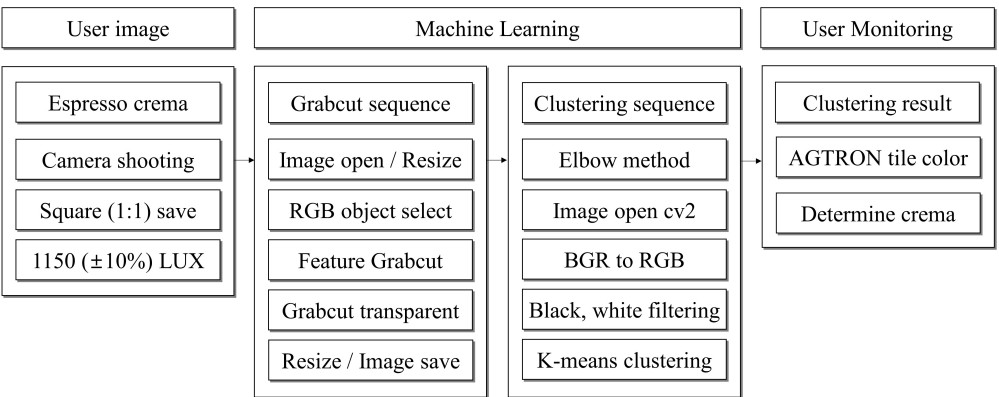

**Figure 2.** Machine learning workflow diagrams used in experiments.

## 2. Related Work

### 2.1. ML in Coffee Prediction

Ahmad et al. [6] applied ANN to develop an automatic coffee sorting system based on image processing. Oliveira et al. [7] used a Bayesian classifier to classify the images of coffee into four colors: white, green, blue green, and cane green. Faridah et al. [10] made a classifier using a neural network algorithm for green coffee samples containing several grains, corresponding to Indonesian standards. Montes et al. [11] performed a segmentation of coffee fruits to classify them according to their maturation state. Suarez et al. [12] showed that the quality of coffee in cups can be predicted as high or low by analyzing the characteristics of green coffee beans using neural networks.

### 2.2. OpenCV Library and GrabCut Algorithm

Open Source Computer Vision (opencv-python 4.7.0.72) [13,14] is an open-source computer vision library that provides a variety of functionalities for image and video processing, computer vision tasks, and machine learning. OpenCV can be utilized in various programming languages such as C++, Python, Java, and more. Its key features include image and video analysis, facial recognition, object detection, and image filtering, among others.

GrabCut [15] is one of the image segmentation techniques used to extract objects from videos. Primarily employed for separating the foreground from the background, it relies on the precise specification of the user's area of interest, allowing the algorithm to perform segmentation accordingly. GrabCut initially models the foreground and background based on the user-specified region, using this model to segment the entire image. It offers high performance through a relatively straightforward implementation. Moreover, image-processing libraries such as OpenCV provide functions implementing the GrabCut algorithm, enabling quick monitoring services through lightweight machine learning analysis.

*2.3. Coffee Roasting Features (Agtron) and Best Espresso Crema*

Roasting refers to the process of applying heat to green coffee beans, during which the beans undergo continuous complex physical and chemical transformations where cellular tissues such as carbohydrates, fats, proteins, and organic acids are broken down and destroyed, resulting in the expression of color, flavor, and aromatic components [9]. While the criteria for roasting may vary among individuals who enjoy coffee beverages, it is observed that specific color differences and distribution occur when extracting coffee using an espresso machine. Furthermore, when an espresso machine is used for extraction at high temperatures and pressures, the fatty and water-soluble components of the coffee blend create a fine golden crema. One characteristic of good crema is the so-called 'tiger skin', where the coffee is uniformly extracted, forming a speckled pattern inside the espresso cup [16]. It is known that when the distinctive color of espresso appears, the best balance of sweetness, bitterness, and acidity is achieved. To analyze the characteristics and form of espresso in real time, we applied the Grabcut and K-means algorithms [17].

## 3. Overview of the Approach

We propose a new analytical approach using unsupervised learning through clustering algorithms to analyze crema extracted from the espresso machine in real time and provide the results to the users. This solution consists of an ML algorithm for crema analysis, as depicted in Figure 2, and a laptop and Jetson Nano computing approach for on-machine operation. In the ML algorithm, our approach to analyzing the crema involves using image segmentation techniques such as Grabcut, which separates an image into foreground and background regions. Clustering algorithms can be applied to group similar regions of crema according to their properties, such as color, texture, and other characteristics. These algorithms can also be used to identify and quantify the amount of crema present in the shot.

*3.1. Grabcut Segmentation*

Grabcut [15] is an image segmentation technique used to separate an object of interest from the background in an image. The algorithm uses a graph-based approach to perform the segmentation, where the image is modeled as a graph, and the segmentation is obtained by finding the minimum cut in the graph.

The mathematical expressions of the GrabCut algorithm are highly technical and intricate. As this paper focuses on the usage of the application, a brief explanation is provided here regarding some key concepts used by GrabCut. It is important to note that these explanations greatly simplify the mathematical details, and, in reality, more complex mathematical formulations are used. Firstly, GrabCut segments the image by minimizing an energy function. The energy function is represented as follows:

$$E(X) = U(X) + V(X)$$

$X$ is a variable representing the segmentation of the image; $U(X)$ corresponds to the data term, representing the similarity between pixels of the foreground and background; and $V(X)$ is the regularization term indicating the similarity between smooth regions.

Secondly, graph cutting is associated with minimizing the energy function. A graph cut is represented using vertices and edges in a graph. In GrabCut, graph cutting is employed to distinguish between the foreground and background:

$$E(X) = \sum_{p \in \text{image}} \left( D_p(X_p) + \lambda \sum_{q \in N(p)} V(X_p, X_q) \right)$$

$D_p(X_p)$ represents the data loss term; $V(X_p, X_q)$ is the regularization term; $\lambda$ is the weight; and $N(p)$ denotes the set of pixels adjacent to pixel p.

*3.2. Espresso Crema and Color Clustering*

To analyze the foam layer of the espresso surface, we applied an unsupervised clustering algorithm [18]. Clustering separates data into a predefined number of clusters by

identifying similarities among data points. Through cluster analysis of espresso, we can establish criteria based on different extraction levels.

$$X = \left\{x^1, x^2, \ldots, x^N\right\} \rightarrow C_1, C_2, \ldots, CM$$
$$\cup_{k=1}^{M} C_k = X\left(\forall i, \forall ji fi \neq j, C_i \cap C_j = \varnothing\right)$$

Similarity measure: This is about evaluating the similarity between a pattern vector x and cluster centroids and establishing rules to assign patterns to specific cluster centers.

$$D = \|x - p\|, D = (x - m)^T \sum{}^{-1}(x - m) \quad (\sum : \text{covariance matrix })$$

Clustering reference function: This is used to evaluate how well a cluster is formed.

$$\sum_{J=1}^{Nc} \sum_{x \in c^j} \|x - m^j\|, \quad m^j = \frac{\sum_{x \in c^j} x}{N_j}$$

The number of clusters is denoted by $N_c$. Each cluster is represented by $C^j$, where $j_{\text{th}}$ refers to the specific cluster. The center vector of the $j_{\text{th}}$ cluster is denoted as $m^j$. $N_j$ represents the number of data points in the $j_{\text{th}}$ cluster.

### 3.2.1. K-Means Algorithm

The K-means algorithm [19] is a clustering algorithm that groups the data given into k groups by minimizing the variance of the distance differences between each group. It works by minimizing the variance of distance differences between each cluster.

The K-means algorithm offers a significant advantage in speed as it calculates distances between points and group centroids for every operation. However, the K-means algorithm comes with several drawbacks. One crucial challenge is the need to determine the appropriate number of groups or classes. When employing clustering algorithms, it is ideal to comprehend the algorithm to gain insights from the data. Think of it as understanding the dynamics of a neighborhood before deciding how to divide it. Unfortunately, the K-means algorithm does not provide inherent understanding, necessitating prior knowledge or additional exploration to select the correct number of clusters.

In essence, while the K-means algorithm excels in rapid calculations, it operates within the constraints of requiring clear instruction regarding the number of clusters and might yield varying outcomes due to its random initialization. Therefore, careful consideration and, potentially, multiple runs are warranted to ensure robust and meaningful results in practical applications. Therefore, in this paper, we leverage the rapid advantages of the K-means algorithm, considering that crema images have four or fewer colors. We opted for an efficient object segmentation method with a relatively simple yet effective image preprocessing approach, similar to Grabcut. In this paper, we use the easy-to-use scikit-learn library instead of using Python 3.8.1 a framework for deep learning.

### 3.2.2. Determining K Value

In this study, we used the elbow method [17] to determine the optimal value of K. The inertia represents the sum of the squared distances between each sample and the centroid of the closest cluster. Featured in Figure 3, the Elbow Method is one of the most popular approaches to determine the optimal value of K. Inertia refers to the sum of the squared distances of the samples from their closest cluster centers. The process involves iterating the values of K from one to n, calculating the distortion values for each value of K and computing the distortion and inertia for each value of K within the given range.

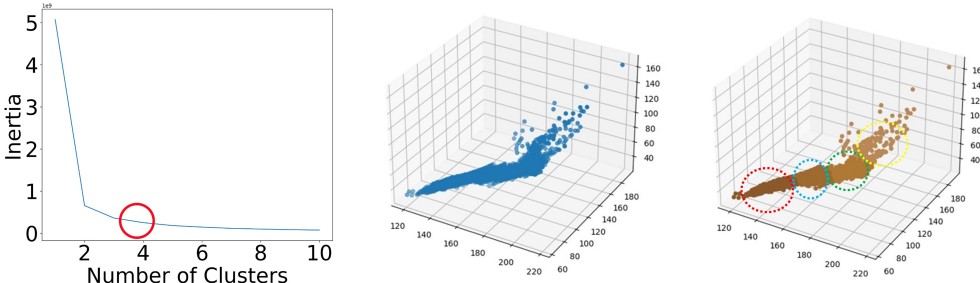

**Figure 3.** Determining the K optimum value of the Elbow method. The multicolored dashed circles represent a visualization of the image array in 3D space (K-means cluster counts).

## 4. Experiment

### 4.1. Overall Design

We aim to establish an automated system using machine learning to analyze crema extracted from actual coffee machines and to provide the results to users, allowing them to adjust the extraction level. When capturing actual espresso photos, as shown in Figure 4, attention should be paid to the surrounding conditions. This is crucial because the crema color of espresso can vary significantly depending on the ambient lighting conditions, including illuminance (LUX) and polarization. Care should be taken to minimize the surrounding shadows and polarization during the photo shoot.

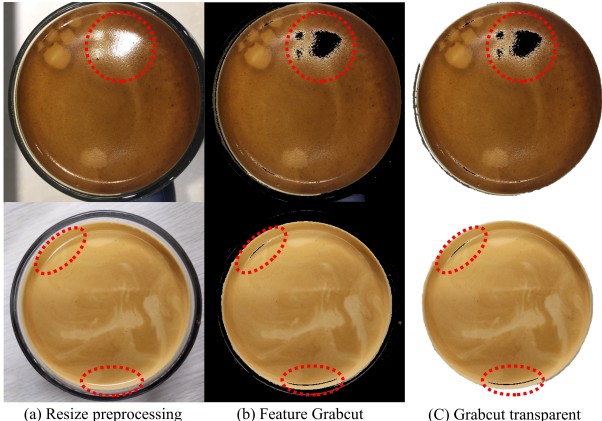

(a) Resize preprocessing　　(b) Feature Grabcut　　(C) Grabcut transparent

**Figure 4.** Traces of polarization remain in the stepwise machine learning process.

We conducted experiments by capturing actual extracted Normal Espresso crema, copying the images to both a laptop and a Jetson Nano, and performing three machine learning algorithms. The first one is basic Color Clustering; the second involves Grabcut, including Color Clustering with instance segmentation; and the third involves our proposed method with Grabcut, Transparent, and Color extraction included in Color Clustering. 1. In the Espresso Crema ML analysis in Figure 5, preprocessing converts high-resolution images captured on mobile devices to a resolution of 500 × 500 pixels for rapid image processing on the device. 2. In Segmentation, Grabcut is used to separate the crema image into foreground and background. Only the crema is cropped and the background is made transparent. 3. In Color Clustering, the image of the separated crema is visualized using the K-means algorithm to show the color proportion for each cluster in terms of RGB values and percentages. 4. In the RGB analysis of crema extraction, the RGB values representing the highest percentage in clustering are highlighted, with the most important RGB color being the Red value within the red square in the RGB analysis. 5. In the Normally Determined Crema, we provided a detailed representation of the RGB values of clustering. Similarly, the red value within the dotted square box is an important variable. In the future, we will compare these results with Agtron color and RGB values to determine the best crema of

espresso. The NVIDIA Jetson Nano and the experimental setup [20] are shown in Figure 6, with the main parameters as follows:

- Coffee Roasting State: Medium, Agtron tile color #55, 10 days (after roasting);
- Espresso extraction amount: a round glass cup, 60 mL (2 shot);
- Crema photography: iPhone 6 square (1:1) save, 1150 ($\pm$10%) LUX;
- Display resolution: Intel UHD Graphics, 1920 $\times$ 1080 32-bit 60 Hz;
- Laptop: Intel Core i7-10510U CPU, 16 GB RAM, Windows 10 Pro, Python 3.8.1;
- NVIDIA Jetson Nano (B01): Quad-core ARM Cortex A57 1.43 GHz, 4 GB 64-bit LPDDR4 25.6 sGB/s, 64 GB microSD, Disable CUDA, Ubuntu 20.04, Python 3.8.1, Manufactured in China as a US product and supplied by MDSTECH company in South of Korea.

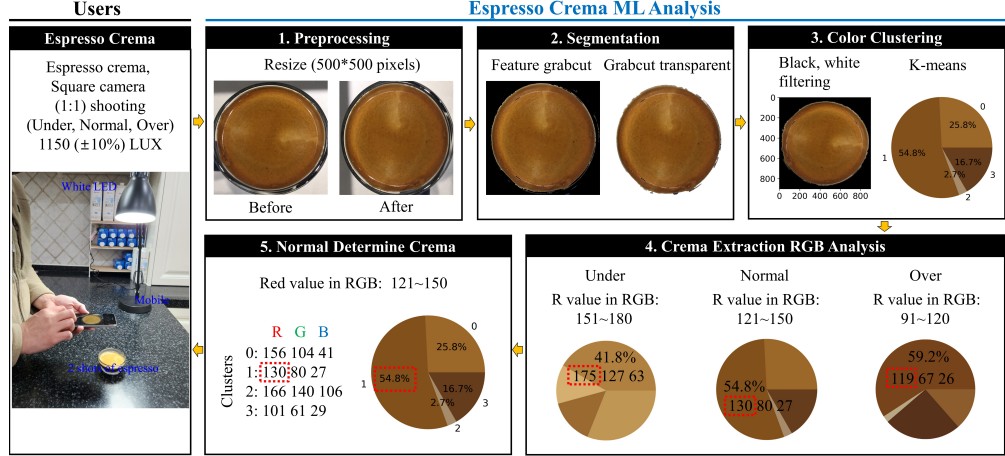

**Figure 5.** User crema shooting and preprocessing, Grabcut, Color Clustering machine learning design.

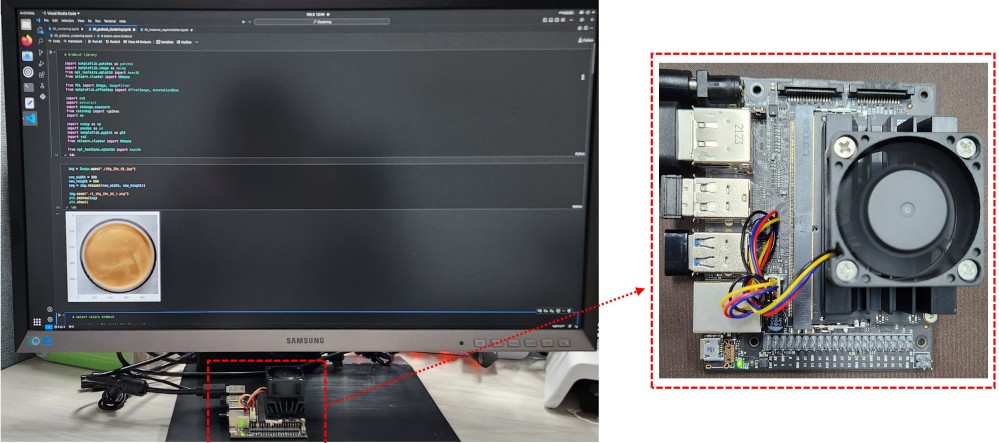

**Figure 6.** NVIDIA Jetson Nano and our experimental setup. (**Left side**) shows our experiment devices. (**Right side**) of the image shows NVIDIA Jetson Nano. We used the Jetson Nano's HDMI port to connect to a monitor. We also provide a detailed video of the laptop used in this experiment and the Jetson Nano running the algorithm (see Appendix C).

### 4.2. Data Preprocessing

The goal of this study was to achieve user-centric lightweight machine learning. Therefore, we assume the machine learning scenario starts from the moment the user captures crema and saves the photo on the device. The objective was to eliminate the need for image editing (cropping) after capturing the photo. However, with most image-processing methods, such as instance segmentation and object detection, when saving and loading images, the OpenCV (CV2) module automatically recreates the external areas of the image even if they are removed, filling them with white or black. This can adversely

affect clustering results. We devoted considerable effort to researching this aspect, as these issues are not easily resolved. By strategically applying color Filtering with Transparent, scikit-learn, and numpy, we addressed these drawbacks, creating an opportunity to provide users with accurate analysis results and automation services.

### 4.2.1. Crema Image Preparation

Preparing the image data plays a crucial role in obtaining accurate results, requiring careful extraction of the image data. However, given the diverse environments of coffee shops and the participation of non-experts in taking photographs, it is necessary to establish standards and provide guidance in this regard. During the experiment, the crema in the espresso extraction cup is shot and saved using a mobile device or a smartphone camera. To ensure minimal distortion and unwanted background, the image is captured in a square format with a (1:1) aspect ratio. The images stored on the Jetson Nano are adjusted to a resolution within the range of 300–500 pixels, considering factors such as memory usage, transmission time, and processing requirements. These adjustments are made before saving the images for further analysis. Through this process, the captured cream images can be appropriately formatted and optimized on the Jetson Nano, enabling an efficient and effective evaluation of cream quality.

As shown in Figure 7, Color Clustering was performed based on the brightness of the lighting. The measurement results indicated that at 1150 LUX and 680 LUX, 96.7% and 97.5%, respectively, of the valid crema color area were determined, with the lighting having a minimal impact on judgment. Furthermore, within coffee shops where the photography was conducted under consistent lighting conditions, discerning the crema's status as Under, Normal, or Over posed no difficulty. However, considering various user environments such as camera angles and lighting conditions, the most crucial factor in capturing crema was found to be the camera angle, which causes streaks, shadows, and reflections, as illustrated in Figure 4. To minimize these potential risks, a fixed tripod cannot be used. Therefore, to address these challenges, we securely placed the 6 cm circular glass used in the experiment for square-shaped capture, reducing camera angle deviations, and maintained a fixed lighting brightness of 1150 (±10%) LUX.

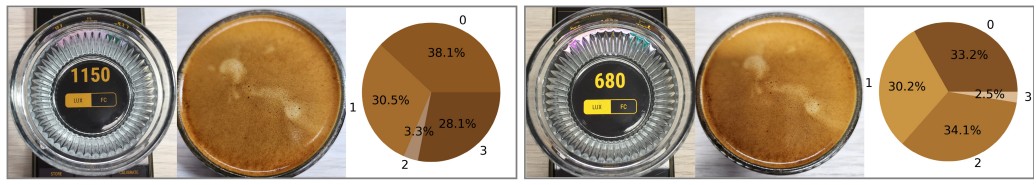

(a) Espresso Crema 1150 LUX Camera Shots  (b) Espresso Crema 680 LUX Camera Shots

**Figure 7.** Using a light meter app on your phone to measure brightness and compare clustering results. (A number of 0-3 means the number of clustering).

### 4.2.2. Automatic Image Processing in Real Image

The crema photos taken and transmitted by the user must go through the appropriate steps for the Color Clustering application. In reality, these captured photos include the colors of the surrounding background and often exhibit light reflections. Failure to effectively remove light reflections and background can result in inaccurate clustering analysis of the cream. Although light reflection and extraction are typically removed during the data preprocessing stage, for automation purposes, the application of efficient algorithms capable of effectively removing them is necessary. Additionally, during the data handling process, the CV2 module of OpenCV automatically treats areas outside the image as white or black, which can affect crema clustering. To address these challenges, we successfully applied the Grabcut algorithm along with the Transparent and Filtering algorithms. This allowed us to extract clustering results based solely on the cream color from the images captured by users. As a result, manual image editing is no longer required, and crema

Color Clustering can be directly performed within the machine, thereby enhancing the user experience.

### 4.3. Evaluation Using Agtron Standard

For evaluation purposes, we adopted the Agtron M-Basic measurement method of the Specialty Coffee Association of America (SCAA). Analyzing the Agtron scale (see Appendix A) of coffee beans by Color Clustering, we regarded deviations in the RGB values of the beans used in the experiment [21] as indications of foreign matter. This enabled us to determine whether the extracted shot was over-extracted or under-extracted. As shown in Table 1, the Agtron M-Basic machine measures color using the Agtron Gourmet Color Scale, which ranges from 0 (black) to 100 (white). The categorized base color values were extracted and evaluated using a graphics editor in this study. For coffee beans, the color classification generally falls into three categories: light, medium, and dark colors [22].

**Table 1.** Characteristics of taste and aroma according to the degree of roasting (*: The beans used in this experiment).

| Roast Classification | Feature of Taste | SCAA Color Tile | Agtron Number | RGB Color |
|---|---|---|---|---|
| Very Light | Pungent and sour | 95 | 80–95 | 206, 129, 1 |
| Light | Strong acidity and subtle body | 85 | 70–80 | 196, 124, 4 |
| Moderately Light | Bright and sour | 75 | 60–70 | 180, 107, 12 |
| Light Medium | Zest | 65 | 60–70 | 153, 85, 21 |
| Medium * | Light acidity, light-bodied | 55 | 50–60 | 139, 75, 27 |
| Moderately Dark | Delicate acidity, full-bodied | 45 | 45–50 | 122, 71, 25 |
| Dark | Strong sweetness, bitterness | 35 | 35–40 | 103, 63, 28 |
| Very Dark | Weak sweetness, burnt taste | 25 | 25–30 | 83, 44, 27 |

Light roast: Beans with Agtron values between 70–85 are considered light roast. These beans tend to be light brown in color and are commonly used to produce milder coffees.

Medium roast: Beans with Agtron values between 55–70 are considered medium roasted. These beans have a darker brown color compared to lightly roasted beans and are commonly used to produce well-balanced flavored coffees. We used beans corresponding to Agtron 55, considering this range as ideal for extraction.

Dark roast: Beans measured with Agtron values below 55 are considered dark roast. These beans have a very dark brown or black color and are typically used to produce bold and intense-flavored coffees.

### 4.4. Experimental Result

4.4.1. Crema Image Result

Our experiment involved testing crema images for various extraction cups, considering various environments (see Appendix B). Figure 8 illustrates the step-by-step crema image transformation process during the machine learning, along with the final clustering results for the entire set of extraction cups. The reason for testing various extraction cups was to evaluate our proposed machine learning algorithm. Ultimately, examining the items in (3) of Figure 8c reveals that the clustering effectively captures the crema color distribution. The most crucial factor in crema images is the persistent presence of black coloration throughout the machine learning process, as shown in Figure 4. This issue is evident in both algorithms

in Figure 8a,b, where the Grabcut transparent phase is successful during the testing phase, but the black color reappears in the final Color Clustering results. By combining our proposed polarization removal method and the effective Grabcut background transparency method, we addressed the drawbacks of OpenCV, resolving black-and-white issues, and applied the algorithm exclusively to the crema area. This approach led to optimized execution of K-means and successful Color Clustering results.

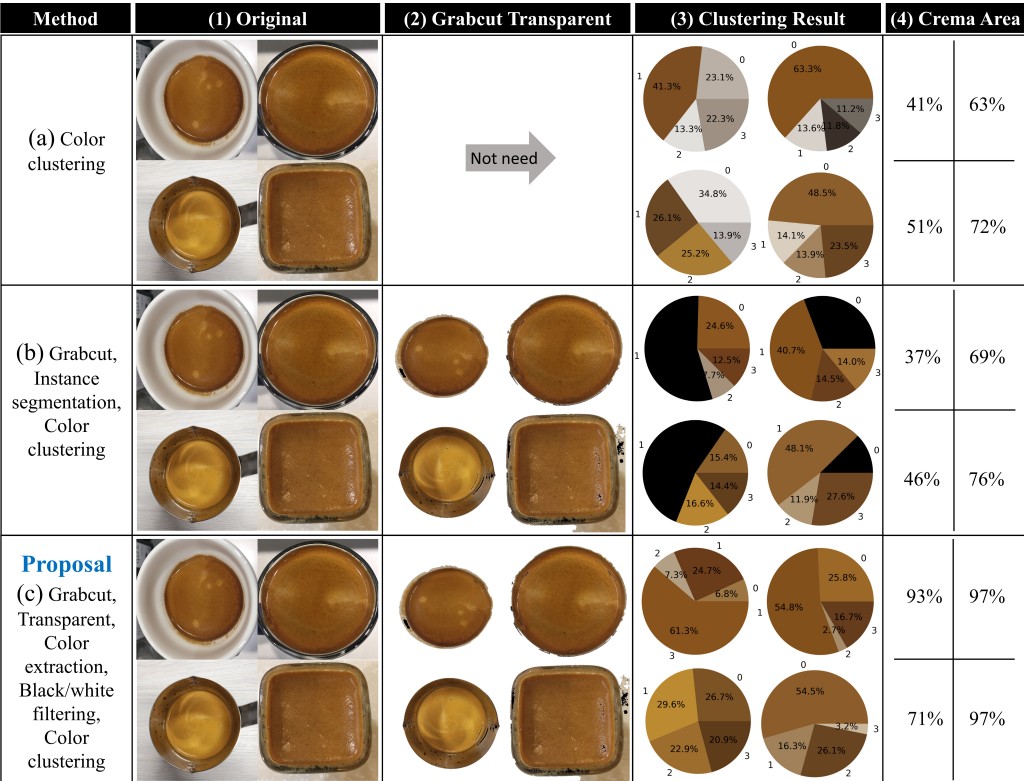

**Figure 8.** Clustering results in different espresso crema normal extraction forms. (**a**,**b**) are the processing for a typical machine learning algorithm, and (**c**) is the machine learning algorithm that we propose by solving all the problems. Numbers (1)–(3) show the algorithm's transformation process from Grabcut to clustering results, and number (4) shows the distribution of crema colors for the results of clustering.

### 4.4.2. Evaluating Cup-Dependent ML Classification Effectiveness and Color Clustering Regions

Figure 8 shows the various cup machine learning results used in this study. Analyzing crema across different cups, we found that image extraction worked best with the round glass cup, which was then utilized for the final crema Color Clustering analysis. To assess how effectively clustering extracted crema images was performed, we categorized the Crema Area item by percentages. Four clusters were classified, combining all brown areas close to the color of crema, excluding white and black with low concentration. As a result, various cups yielded crema area results ranging from 71% to 97% in (4) of Figure 8c. The Crema Area, with the round glass cup used in the experiment, shows a Color Clustering result of 97%. These findings indicate that our proposed algorithm, which uses only the crema color from the original images through automated Grabcut, Filtering, and Transparent, while removing irrelevant color proportions, can achieve a high crema area ratio. Additionally, since the proposed algorithm effectively utilizes crema color, consistently high crema areas can be obtained across all cups.

Examining the general criteria for proper espresso crema extraction based on the aforementioned results, typical standards include between 9 and 10 bars of pressure, a water temperature ranging from 90 to 95 °C, 18 to 20 g of finely ground coffee in the portafilter

basket, and an extraction time from 20 to 30 s. Deviating from these standards can lead to abnormal extraction of espresso crema, as illustrated in Figure 9. In optimal extraction, most Agtron color values (see Appendix A) concentrate around 45, while under-extraction shows significantly lighter colors with values above 65. Conversely, over-extraction results in a predominance of color values below 35. Through these observations, it is evident that the Color Clustering algorithm can identify subtle differences perceptible to the naked eye with high precision. Notably, in cases of over-extraction, dark and light colors in the direction of white are observed within the red dashed line. This phenomenon occurs as the extraction time increases, resulting in darker colors being extracted. Furthermore, once all components of the coffee bean are extracted, only lighter colors remain.

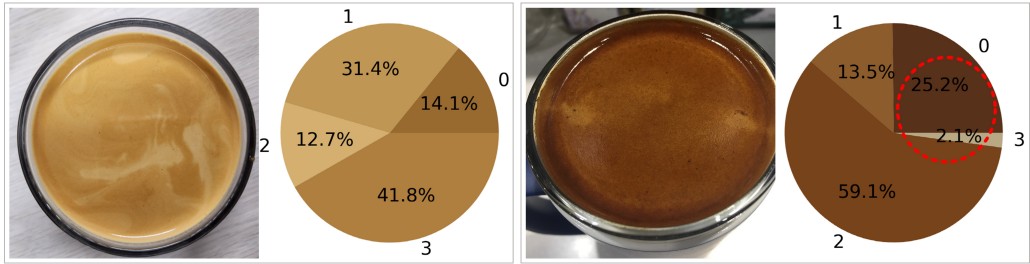

**Figure 9.** Clustering result of abnormal extraction. (A number of 0–3 means the number of clustering).

### 4.4.3. ML Real Time with Jetson Nano and Laptop

Since our primary goal is monitoring services, we measured the time up to visualization, using the Jupyter Notebook package within Visual Studio Code, widely employed for Python visualization. We measured the time it takes to generate images that end-users can inspect. Table 2 displays the times measured using the Jupyter Notebook. Although the displayed time may not appear in real time, improvements in hardware performance correspond to faster processing speeds. Moreover, we designed our experiment to be lightweight, enabling integration with the IoT from the outset. With a free cloud service update time (sensor data) of 10–15 s, the time is sufficiently realistic for users to utilize. Table 2 (*) shows the efficient machine learning algorithm methods that we have proposed.

**Table 2.** Runtime measured using Visual Studio Code (VSCode) and Jupyter Notebook on Jetson Nano and laptop. (*: The machine learning algorithm that we propose to solve all problems) We provide a video of the algorithm execution time on the laptop and Jetson Nano used in this experiment.

| Python | | Jetson Nano | Laptop |
|---|---|---|---|
| Module Import | | 6.7 s | 2.7 s |
| Machine Learning | Color Clustering | 15.7 s | 2.2 s |
| | Grabcut, Instance segmentation | 22.1 s | 10.0 s |
| | Grabcut, Transparent, Color extraction * | 11.6 s | 6.1 s |

### 4.4.4. Determining Extraction Crema

In Python programming, for K-means clustering, there's an RGB image of size M × N with pixel intensities. This image consists of M × N pixels, each containing three components: red, green, and blue. When treating these M × N pixels as data points, K-means clustering is applied. Pixels belonging to a specific cluster have colors more similar to each other than to pixels in other clusters. For this reason, when evaluating the results of espresso crema extraction, as shown in Figure 10, the RGB red component and the Agtron tile color are compared, using the characteristics of K-means without using the clustering results of the green and blue values.



| Crema Extraction | Color clustering RGB result (Clusters = 4) | | Agtron Evaluation Color Red Value | Grinding / Extraction Time | Lighting (LUX) |
|---|---|---|---|---|---|
| Under |  | R  G  B<br>0: [175, 127, 63]<br>1: [212, 175, 111]<br>2: [153, 106, 48]<br>3: [192, 150, 84] | R: 151~180<br><br>65 | 16 g / 16 s | 1150~1160 |
| Normal |  | R  G  B<br>0: [156, 104, 41]<br>1: [**130**, 80, 27]<br>2: [166, 140, 106]<br>3: [101, 61, 29] | R: 121~150<br><br>45 | **20 g / 22 s**<br><br>Normal range<br>(20~30 s) | 1150~1160 |
| Over |  | R  G  B<br>0: [119, 67, 26]<br>1: [185, 168, 132]<br>2: [88, 49, 27]<br>3: [140, 92, 45] | R: 91~120<br><br>35 | 22 g / 32 s | 1150~1160 |

**Figure 10.** Determining the best espresso Crema.

Using a single clustering, accurate results can be obtained when compared with the Agtron color. The reason for conducting four clusters in this study is to target advanced users. Crema tends to have a light color when the extraction time is maintained at 30 s or more, indicating that the coffee bean components have been extracted. This is why we conducted four clusterings to identify bright areas, comprising 2.1–2.7% of the clusters in the "Normal" and "Over" categories in Figure 10.

In the RGB results of the color clustering for the "Normal" category, if the RGB of one cluster [130 80 27] constitutes more than 50%, and if this cluster falls within the Agtron color values of 121–150, it is considered a proper extraction. Similarly, the extraction results for the "Under" and "Over" categories are assessed in the same manner. As discussed above, "Grinding/extraction time" and "Lighting (LUX)" are aspects of the extraction environment settings used in this experiment. In summary, using the method mentioned in Note 1 for optimal espresso extraction conditions [21] using beans with an Agtron color of 55, the crema appears with a Color Clustering result showing a one-step darker Agtron (see Appendix A) color value, from 55 to 45. Confirmation that the result for the single-cluster RGB [130, 80, 27] falls within the range of Agtron numbers 45 RGB [122, 71, 25] and 55 RGB [139, 75, 27] verifies the evaluation of optimal espresso crema.

## 5. Conclusions

The research paper focuses on achieving desired espresso extraction by applying lightweight and compact algorithms that can run on a device. The approach uses crema photos to predict the extraction level, replacing subjective judgment with objective criteria. Image-processing techniques, such as Grabcut and transparency transformation, are implemented to automate the algorithm within the coffee machine. These techniques enable automatic cropping of the coffee cup region, extraction of the final crema image, and removal of unnecessary colors. The measured crema is compared to Agtron color tile values for evaluation. Furthermore, the article highlights the influence of various factors, such as coffee beans, degree of roasting, grinding level, extraction time, and temperature, on the taste and aroma of coffee. By automating these factors, in addition to controlling particle size, the research demonstrates a successful fully automated process for extracting optimal crema within the coffee machine.

Discussion for future coffee machines: The research paper discusses the existence of optimal extraction conditions in espresso extraction, as well as three representative espresso extraction standards: "Andrea Illy & Rian Van Den Oord's Viani", "WBC", and "Instaurator". These standards share common parameters, including water pressure (9–10 bar), water temperature (90–95 °C), portafilter basket capacity (18–20 g), and extraction time (20–30 s). When under-extraction or over-extraction is identified, adjustments in grind size and time using a grinder are necessary. The proposed algorithm uses espresso crema to help users assess the level of espresso extraction. This approach demonstrates the potential for further advances in the field. The taste and aroma of coffee are influenced by various factors such as coffee beans, roasting degree, grinding level, extraction time, and temperature. Additionally, the integration of espresso crema and coffee machine sensor data allows for consistent post-processing management. In summary, the article emphasizes the importance of optimal extraction conditions in espresso extraction and introduces three representative extraction standards. The proposed algorithm, utilizing espresso crema, aids in evaluating the extraction level of espresso. The research showcases the potential for future advancements and highlights the ability to control various factors using the IoT and Agtron color tile values. Furthermore, it discusses the establishment of a unified environment by integrating espresso crema and coffee machine sensor data for consistent post-processing management.

**Supplementary Materials:** The following supporting information can be downloaded at: https://www.mdpi.com/article/10.3390/electronics13040800/s1, Video S1: electronics-2843492-supplementary.

**Author Contributions:** Conceptualization, J.C.; methodology, J.C.; software, J.C.; validation, J.C. and K.K.; formal analysis, J.C.; investigation, J.C. and S.L.; resources, J.C. and K.K.; data curation, J.C.; writing—original draft preparation, J.C.; writing—review and editing, S.L., H.S. and K.K.; visualization, J.C.; supervision, K.K.; project administration, H.S.; funding acquisition, K.K. All authors have read and agreed to the published version of the manuscript.

**Funding:** This research was supported by the MSIT (Ministry of Science and ICT), Korea, under the Grand Information Technology Research Center support program (IITP-2024-2020-0-01741) supervised by the IITP (Institute of Information & Communications Technology Planning & Evaluation). This work was supported by the Catholic University of Korea, Research Fund, 2022 (No. M2022-B0008-00145).

**Institutional Review Board Statement:** Not applicable.

**Informed Consent Statement:** Not applicable.

**Data Availability Statement:** Data are contained within the article.

**Conflicts of Interest:** The authors declare no conflict of interest.

## Abbreviations

The following abbreviations are used in this manuscript:

| | |
|---|---|
| ML | Machine Learning |
| IoT | Internet of Things |

## Appendix A

The taste and aroma of coffee vary distinctly depending on the degree of roasting, so it is essential to determine the roasting point in line with the characteristics of the coffee. The color steps used by Agtron M-basic represent a classification of 0 to 100, and these color steps were extracted and converted to RGB, C++ and Web colors using a graphical editor.

| Roasting stage | Coffee taste characteristics | Agtron M-basic | Color |
|---|---|---|---|
| **Very Light** | It has a light brown color, a pronounced acidity, and a mild aroma and body. This refers to a very lightly. | 95 | RGB 206 129 1<br>C++ 0x181CE<br>Web #CE8101 |
| **Light** | It has a strong acidity, and the surface of the coffee beans is dry, allowing for the detection of any shortcomings in the coffee. | 85 | RGB 196 124 4<br>C++ 0x47CC4<br>Web #C47C04 |
| **Moderatery Light** | It delivers a noticeable acidity and exhibits a nutty flavor. | 75 | RGB 180 107 12<br>C++ 0xC6BB4<br>Web #B46B0C |
| **Light Medium** | The acidity is slightly mild, marking the point where the body begins to emerge. | 65 | RGB 153 85 21<br>C++ 0x155599<br>Web #995515 |
| **Medium** | The acidity nearly disappears, revealing more distinct characteristics of the coffee bean variety. | 55 | RGB 139 75 27<br>C++ 0x1B4B8B<br>Web #8B4B1B |
| **Moderatery Dark** | The sweetness intensifies, and the coffee bean surface reflects oils. | 45 | RGB 122 71 25<br>C++ 0x19477A<br>Web #7A4719 |
| **Dark** | The characteristics of the variety diminish, and a strong sense of body becomes prominent. | 35 | RGB 103 63 28<br>C++ 0x1C3F67<br>Web #673F1C |
| **Very Dark** | The body and sweetness diminish, and a stronger bitterness emerges. | 25 | RGB **83** 44 27<br>C++ 0x1B2C53<br>Web #532C1B |

**Figure A1.** Roasting stage and Agtron number of M-Basic color tile.

**Appendix B**

Images taken for Color Clustering analysis of the various cups and crema used in this experiment with extraction complete. To consider the different environments in (a)–(d), we conducted experiments with different espresso cups. The results of the clustering algorithm for different cups are shown in Figure 8. The espresso crema cup used for the final analysis of Under, Normal, and Over in this experiment is (b).

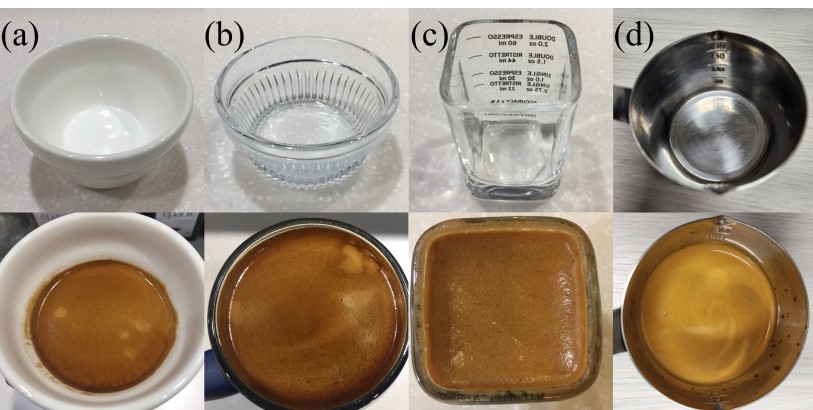

**Figure A2.** Extracted results and cups in experiments.

**Appendix C**

The text presents real-time program execution speeds based on the results obtained from Figure 6 and Table 2, similar to the NVIDIA Jetson Nano experimental environment. A video captured during the experiment is provided. The experimental setup involved connecting a display via the HDMI port and using a wireless keyboard and mouse to run the program. Visualization and measurement time monitoring were carried out using Python's Jupyter Notebook (Jupyter Core 4.9.1), which is licensed under an open-source license (Supplementary Materials).

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
