# Peer review of "Lightweight Machine Learning Method for Real-Time Espresso Analysis"

_electronics, doi:10.3390/electronics13040800_

Round 1
Reviewer 1 Report
Comments and Suggestions for Authors
The authors present a novel and easy-to-use automated visual assessment technique for measuring the quality of Espresso. Espresso is one of the most widely consumed beverages in the world, accounting for a significant share of the coffee market, which makes this manuscript pertinent to the current scenario. However, this manuscript lacks a clear and convincing explanation of the method used, and the evidence presented from the experiments is not sufficient to justify the main conclusions. Therefore, this manuscript requires substantial revisions to enhance its quality.
1. Espresso quality is measured by a variety of indicators, such as color, aroma, and taste, among others. However, the connection between visual attributes and Espresso quality is not well established (SCAA Quality Score). The introduction section does not adequately explain how the visual features of Espresso influence its quality, and it lacks sufficient and credible evidence to support this claim.
2. The authors do not provide a clear rationale for selecting Grabcut and K-means among other well-known methods in the introduction or relevant literature review sections.
3. The method section does not provide a detailed description of the Grabcut and Clustering methods, nor does it explain how the final evaluation metric is derived from these methods. I believe this is a crucial aspect of this manuscript that needs to be clarified.
4. Using a handheld phone to take photos results in inconsistent images (Figure 3. “Users” part), and a tripod is recommended for better stability.
5. I think that the experimental outcomes are highly dependent on the camera angle and lighting conditions, etc. Therefore, it is essential to address how these factors can be controlled or minimized for the proposed method to be applicable in practice. However, the authors did not address this issue in their experimental design or analysis, nor did they examine how varying lighting conditions affect the results.
Reviewer 2 Report
Comments and Suggestions for Authors
This paper combined Crabcut and clustering algorithms to evaluate coffee crema and demonstrates the effectiveness of the approach through experiments. The main issues in the paper are as follows:
1、The title of the article is "Lightweight Machine Learning Method for Real-Time Espresso Analysis", but real-time processing is not reflected in the experiments. In my opinion, the experiment should highlight the algorithm's processing speed. Additionally, it is recommended to include more relevant introduction to lightweight methods in Section I, such as:
[1] Sun, Z.; Leng, X.; Lei, Y.; Xiong, B.; Ji, K.; Kuang, G. BiFA-YOLO: A Novel YOLO-Based Method for Arbitrary-Oriented Ship Detection in High-Resolution SAR Images. Remote Sens. 2021, 13, 4209. DOI: 10.3390/rs13214209.
[2] X. Ma, K. Ji, B. Xiong, L. Zhang, S. Feng and G. Kuang. Light-YOLOv4: An Edge-Device Oriented Target Detection Method for Remote Sensing Images [J]. IEEE Journal of Selected Topics in Applied Earth Observations and Remote Sensing, vol. 14, pp. 10808-10820, 2021. DOI: 10.1109/JSTARS.2021.3120009.
2. In Section I, one of the contributions mentioned in the paper is the "Intelligent automatic crema extraction system in the coffee machine through embedded computing", but only Jetson Nano is mentioned in Sections III and IV. It is necessary to provide more information and include the running speed of embedded devices in the experiments.
3. In Section III, the main work should be emphasized instead of directly concatenating existing methods.
Comments on the Quality of English LanguageModerate editing of English language required.
Reviewer 3 Report
Comments and Suggestions for Authors
Review of “Lightweight Machine Learning Method for Real-Time Espresso Analysis”
This work talks about a lightweight algorithm for real-time analysis of coffee crema extraction. Embedded computing is incorporated which helps in providing end-to-end implementation. Though there are multiple works leveraging mostly image processing for coffee sorting, the embedded application and machine monitoring service adds uniqueness to this work. Some inputs:
· Any specific advantage of using Grabcut image segmentation technique?
· Given the impact of ambient lighting condition on this technique, it will be good to add some statical significance of the data captured through error bars or something equivalence.
· It will be good to add some sample snapshots highlighting the data processing step of color filtering or a supplementary video showing the real-time implementation on real image to understand the timescale.
· Some validation of the results from color clustering will be important to add.
· The embedded and real time application is not clear- also the value addition of the works can be further enhanced. Here, mostly post-espresso extraction, the technique is used to understand the characteristics. Can we use this technique to predict some of the control parameters that leads to desired espresso extraction?
· Several factors are mentioned which impacts the taste and aroma of coffee- can those be quantified and introduced in the ML model for prediction purposes?
Author Response
첨부파일을 참조하시기 바랍니다.

Round 2
Reviewer 1 Report
Comments and Suggestions for Authors
All my concerns have been properly addressed, , and I think this manuscript is ready to be published.
Reviewer 2 Report
Comments and Suggestions for Authors
The authors have addressed all my concerns. For my side, it's ready for publication.
Comments on the Quality of English LanguageMinor editing of English language required.
Reviewer 3 Report
Comments and Suggestions for Authors
The article looks fine now